# The Combined Use of 5-ALA and Chlorin e6 Photosensitizers for Fluorescence-Guided Resection and Photodynamic Therapy under Neurophysiological Control for Recurrent Glioblastoma in the Functional Motor Area after Ineffective Use of 5-ALA: Preliminary Results

**DOI:** 10.3390/bioengineering9030104

**Published:** 2022-03-03

**Authors:** Elizaveta I. Kozlikina, Igor S. Trifonov, Mikhail V. Sinkin, Vladimir V. Krylov, Victor B. Loschenov

**Affiliations:** 1Prokhorov General Physics Institute of the Russian Academy of Sciences, 119991 Moscow, Russia; loschenov@nsc.gpi.ru; 2Institute for Physics and Engineering in Biomedicine, National Research Nuclear University MEPhI, 115409 Moscow, Russia; 3Department of Neurosurgery, Federal State Budgetary Educational Institution of Higher Education “A.I. Evdokimov Moscow State University of Medicine and Dentistry”, The Ministry of Healthcare of the Russian Federation, 127473 Moscow, Russia; dr.trifonov@mail.ru (I.S.T.); mvsinkin@gmail.com (M.V.S.); krylov@neurosklif.ru (V.V.K.)

**Keywords:** 5-ALA, chlorin e6, fluorescence diagnostics, photodynamic therapy, glioblastoma, neurophysiological control

## Abstract

The treatment of glial brain tumors is an unresolved problem in neurooncology, and all existing methods (tumor resection, chemotherapy, radiotherapy, radiosurgery, fluorescence diagnostics, photodynamic therapy, etc.) are directed toward increasing progression-free survival for patients. Fluorescence diagnostics and photodynamic therapy are promising methods for achieving gross total resection and additional treatment of residual parts of the tumor. However, sometimes the use of one photosensitizer for photodynamic therapy does not help, and the time until tumor relapse barely increases. This translational case report describes the preliminary results of the first combined use of 5-ALA and chlorin e6 photosensitizers for fluorescence-guided resection and photodynamic therapy of glioblastoma, which allowed us to perform total resection of tumor tissue according to magnetic resonance and computed tomography images, remove additional tissue with increased fluorescence intensity without neurophysiological consequences, and perform additional therapy. Two months after surgery, no recurrent tumor and no contrast uptake in the tumor bed were detected. Additionally, the patient had ischemic changes in the access zone and along the periphery and cystic-glial changes in the left parietal lobe.

## 1. Introduction

Glial brain tumors have diffuse and infiltrative growth patterns, so achieving their complete resection is often complicated [1,2]. The treatment of malignant brain gliomas is one of the most challenging tasks in neurooncology. Currently, combined methods of treatment are a priority because monotherapy does not lead to significant improvements in immediate and long-term results. Combined treatments can include surgical removal of the tumor, radiation, and chemotherapy [3,4], and new methods of treatment, such as specific antitumor immunotherapy [5], fluorescence-guided resection (FGR) [6,7], and photodynamic therapy (PDT) using modern photosensitizers (PSs) [8,9,10]. The prerequisites for the latter’s use as an alternative method of therapy for malignant brain neoplasms were the results of clinical studies that showed significant increases in the median overall survival of patients and the relapse-free period, while reducing the risk of developing severe neurological complications [11].

Since gross total resection is accompanied by better outcomes [2], fluorescence diagnostics (FD) is of particular interest. FD is a fast, intraoperative, and noninvasive method for the detection of tumor boundaries and residual parts [12,13,14]. This method is based on the use of PS, which selectively accumulates in tumor tissues and fluoresces under laser light irradiation at a specific wavelength.

At the moment, the standard PS for FD of gliomas is 5-ALA-induced protoporphyrin IX (PpIX) which has a cellular mechanism of accumulation [10,15]. One of the main advantages of PpIX compared to other PSs is its high level of accumulation in tumor cells [16]. It was shown that the use of 5-ALA helps to achieve a high tumor resection rate and improve patient prognosis [17]. Additionally, in several studies, chlorin e6 (Ce6) and talaporfin sodium were used for FD [18,19].

PSs used for FD can also be applied for controlled PDT to residual parts of a tumor that are impossible to resect [12,20]. At the moment, PDT for glial brain tumors was performed only with one PS, such as PpIX [21] or talaporfin sodium [9,20]. However, this treatment does not work well in all cases.

To improve the clinical outcome of FD and PDT, the combination of two PSs, Ce6 and PpIX, can be used for navigation, necrosis, and/or apoptosis of tumor cells and destruction or thrombosis of the tumor vascular system [22].

The aim of this case report is to describe the preliminary results of the first combined use of two different PSs for FD and PDT under neurophysiological control, including the assessment of the degree of PSs accumulation, and clinical outcomes of the patient with a glioblastoma.

## 2. Case Description

This study presents a 29-year-old male patient. In February 2021, the patient identified involuntary movements of the first finger of the right hand. Magnetic resonance imaging (MRI), functional MRI (fMRI), computed tomography (CT), and CT angiography of the brain were performed and indicated a glial tumor of the left parietal lobe with involvement of the left pyramidal tract in the pathological process (Figure 1a–f). 

### 2.1. First Intervention

The first surgery was in April 2021. Due to the location of the tumor in the motor zone, the operation was performed awake. Tumor resection under neurophysiological monitoring was done with motor mapping involving bipolar electrical cortical stimulation and clinical assessment of patient’s movements. Resection was accompanied by a neuro- (Medtronic StealthStation EM Navigation, Medtronic, Minneapolis, MN, USA) and 5-ALA-navigation using the microscope equipped with a fluorescent 405 nm light module (Carl Zeiss OPMI Pentero 900, Carl Zeiss, Oberkochen, Germany). According to post-surgical CT on the first day after surgery, the tumor was totally resected (Figure 1g–i). However, during resection of the residual parts with increased fluorescence, the patient noted numbness in the 5th finger of the right hand, and later, numbness of the 4th finger. Resection of residual parts was stopped by the patient’s decision due to the high risk of postoperative consequences. After the surgery, monoplegia of the right hand was developed and regressed on the third day.

After histological analysis of the biopsy sample, glioblastoma (grade IV) of the left parietal lobe was confirmed.

Postoperative chemotherapy (temozolomide) and radiotherapy (30 sessions with a focal dose of 2 Gy, total focal dose of 60 Gy) were done.

### 2.2. Second Intervention

In June 2021, the patient noted arm weakness, identifying four points of paresis of the right hand. MRI showed a recurrence of glioblastoma of the left parietal lobe (Figure 2a,b). The second surgery was performed in August 2021 with total tumor resection under fluorescent and neurophysiological monitoring (Figure 2c,d). 5-ALA was sublingually administered to the patient at a 20 mg/kg concentration 4–4.5 h before FD. During FGR, fluorescence spectra of PpIX accumulated in the tumor tissues were recorded using fiber-optic spectrometer LESA (Biospec, Moscow, Russia), He-Ne laser source, and diagnostic optical fiber illumination (Figure 3a). To determine accumulated PpIX concentration in tumor tissue, optical phantoms were prepared: PpIX in various concentrations (1, 5, 10, 15 mg/kg) was mixed with a scattering medium (1% MLT/LCT Intralipid) and poured into test tubes. Fluorescence spectra of optical phantoms were recorded (Figure 3b).

PDT of residual parts with increased fluorescence was performed using a 635 nm laser source (Biospec, Moscow, Russia) and optical fiber illumination with an energy dose of 30 J/cm^2^. The irradiation time was 6.5 min. After the laser irradiation, repeated FD was performed.

In the postoperative period, the right-side upper monoparesis progressed to three points, more pronounced in the wrist, which regressed to four-point paresis in 4 weeks. His Karnofsky performance status scale score was 90.

Immunohistochemical analysis showed focal positive expression of GFAP, residual expression of NF, and negative expression of EMA and NFkp65. No codelets of 1p\19q were detected during the molecular genetic study. The postoperative six-cycle course of chemotherapy with Irinotecan (S.C. SINDAN-PHARMA S.R.L., Bucharest, Romania) and Bevacizumap (Roche Diagnostics GmbH, Mannheim, Germany) was performed.

### 2.3. Third Intervention

In November 2021 the patient developed three-point right-side hemiparesis and one-point paresis of the right wrist. His Karnofsky performance status scale score became 60. Recurrence of the tumor of the left parietal lobe with dimensions of 9 mm × 3 mm × 9 mm, actively accumulating a contrast agent, and massive perifocal edema of the left parietal lobe were detected on MRI (Figure 4a–e).

Four months after the second surgery, a third intervention was performed. 5-ALA (NIOPIK, Moscow, Russia) was sublingually administered to the patient at a concentration of 20 mg/kg 4–4.5 h before the FD. Ce6 (DEKO Company, Moscow, Russia) diluted in 100 mL of saline was intravenously administered to the patients at a 1 mg/kg concentration 3–3.5 h before FD.

Resection of the tumor was carried out under intraoperative neurophysiological monitoring. To determine the zone of safe encephalotomy, cerebral cortex mapping and monitoring of the corticospinal tract integrity using the method of motor evoked potentials were performed. Subcortical mapping using a suction probe was performed throughout the entire operation. When motor evoked potentials appeared with a 5 mA current, the resection was stopped. After the resection of the main tumor and part of the left parietal lobe, sensitized areas with increased Ce6 and PpIX fluorescence were recorded using a fiber-optic spectrometer, He-Ne laser source, and diagnostic optical fiber illumination (Figure 5a). Additional resection of detected areas was performed, and PDT of residual parts with the total energy dose of 60 J/cm^2^ was performed using a 660 nm laser source (Biospec, Moscow, Russia) and optical fiber illumination. The irradiation time was 14 min. After the PDT, repeated FD of the tumor bed was carried out, and no increased fluorescence from either PS was detected (Figure 5a).

To determine accumulated PpIX and Ce6 concentrations in tumor tissues, optical phantoms were prepared with concentrations 1, 5, 7, and 10 mg/kg; and 0.5, 1, 5, and 10 mg/kg, respectively (Figure 5a). The fluorescence spectrum registered before PDT was linearly decomposed into the fluorescence spectra of Ce6 and PpIX optical phantoms, and fluorescence indices were calculated for areas under spectra (Figure 5b).

After the third surgery, the three-point right-sided hemiparesis and one-point paresis of the wrist regressed to three points in the leg and four points in all hand. Ischemic changes in the access zone and along the periphery with a volume of 47 × 28 × 38 mm and cystic–glial changes in the left parietal lobe were observed. CT did not show a recurrent tumor or contrast uptake in the tumor bed (Figure 4f–j).

At the moment, two months after the third surgery, the patient is active within his apartment, moving with the help of crutches. His Karnofsky performance status scale score is 70.

## 3. Discussion

The completeness of glial brain tumor resection is often complicated due to the inaccurate determination of tumor borders and infiltrative tumor growth patterns. Additionally, the detection of functionally significant zones during resection is one of the main criteria for a positive neurological outcome.

Currently, the standard method to achieve a more complete resection is intraoperative fluorescent monitoring using a surgical microscope equipped with a fluorescence module (405 nm excitation wavelength) and 5-ALA [7,23]. However, the light penetration depth into the tissues for this wavelength is about 2–3 mm, which is not enough for a high rate of tumor resection. To overcome this problem, light sources with wavelengths outside the visible spectrum and spectroscopic techniques began to be used in neurosurgical practice for FD—for instance, light with a penetration depth of 7–13.5 mm using a 635 nm laser source [10,20]. Additionally, following PDT with a 5-ALA as a part of combined treatment has been actively used in neurosurgical practice [9,21]. 

According to the literature, the use of only 5-ALA for FD and PDT is generally enough to increase the time of progression-free survival for glioblastoma up to 12–14.5 months, and in patients harboring high-grade gliomas, 13% become long-term survivors (>2 years) after PDT [24,25]. Our case showed achievement of gross total resection of the tumor under fluorescence monitoring according to MRI, but a poor result of PDT with 5-ALA mono use and additional therapy. The time to tumor relapse was 3 months. 

Since the use of only 5-ALA for PDT had not shown a good result for this patient during the second intervention, we proposed that use of two PSs with different mechanisms of accumulation and action during PDT would show a better result. Additionally, the use of two PSs may enhance the quality of FD during tumor resection. During the surgery, we noticed different accumulation pictures for PpIX and Ce6: the maximum accumulation of Ce6 was detected in vessels and the surrounding tissues, whereas PpIX was uniformly accumulated in tumor tissues. However, according to the study [26], Ce6 accumulates not only in the vascular system but also in tumor cells. At a cellular level, Ce6 accumulates primarily in lysosomes and mitochondria. For that reason, Ce6 may work not only by photodamaging the vascular system, but also by promoting an apoptotic response in tumor cells. Therefore, the use of Ce6 and 5-ALA for PDT can lead to both necrosis and/or apoptosis of tumor cells and the destruction or thrombosis of the tumor’s vascular system.

During the third intervention, a 660 nm laser source was used for PDT. Ce6 and 5-ALA have different absorptions at this wavelength (Figure 5c,d). While Ce6 has a significant absorption peak at 660 nm, PpIX mainly excites at 405 and 635 nm. However, taking into account that the penetration depth for 635 nm is significantly lower than for 660 nm, it was shown [27] that the lower absorption of 660 nm by PpIX could be compensated by increased irradiation time. Hence, it may be concluded that the main effect of the third intervention came from Ce6-induced photodamage. However, the effect of PpIX during PDT cannot be dismissed.

## 4. Conclusions

This case report shows the preliminary results of the first clinical use of two PSs for FD and PDT of glioblastoma in one patient after the poor result of PDT with the use of only 5-ALA. During FGR, we detected accumulations of Ce6 and PpIX in tumor tissues in high concentrations. That allowed us to perform total resection of tumor tissue according to MRI and CT images and remove additional tissue with increased fluorescence intensity without irreversible neurological deficit for the patient. Two months after the PDT with Ce6 and 5-ALA, CT showed ischemic changes in the access zone, and along the periphery, cystic-glial changes in the left parietal lobe, which were not detected after PDT with only 5-ALA at the same period of time after interventions. Additionally, no recurrent tumor and no contrast uptake in the tumor bed were detected. Despite the fact that this report represents only one clinical case of a patient with glioblastoma, the use of two PSs for FD and PDT of high-grade gliomas is promising.

## Figures and Tables

**Figure 1 bioengineering-09-00104-f001:**
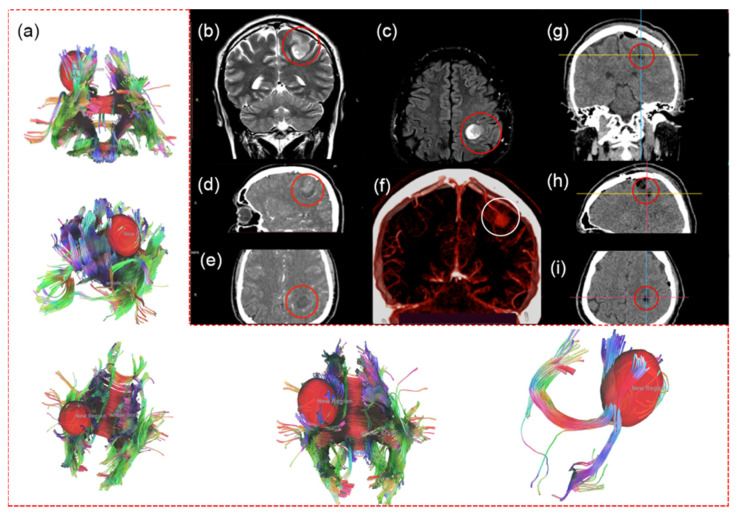
MRI, fMRI, and CT before first surgery. fMRI with involvement of the left pyramidal tract in the pathological process (**a**). MRI (1.5 T) of the tumor of the left parietal lobe (red circle): T2 coronal (**b**), T2 flair axial (**c**) projections. CT with contrast enhancement: tumor of the left parietal lobe with contrast uptake (red circles): sagittal (**d**), axial (**e**) projections. CT angiography (**f**): emissary vein to sagittal sinus (white circle). CT images on the first day after the first surgery: coronal (**g**), sagittal (**h**), and axial (**i**) projections. Total resection of the tumor in the left parietal lobe, no tumor, and no contrast uptake in the tumor bed.

**Figure 2 bioengineering-09-00104-f002:**
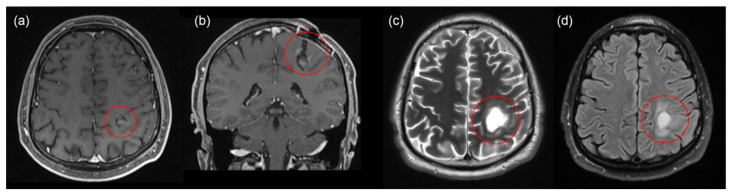
MRI 1.5 T with contrast before and after the second surgery. Before surgery: T1 axial (**a**), coronal (**b**) projections. After surgery: T2 axial (**c**), T2 flair axial (**d**). No recurrent tumor, and no contrast uptake in the tumor bed.

**Figure 3 bioengineering-09-00104-f003:**
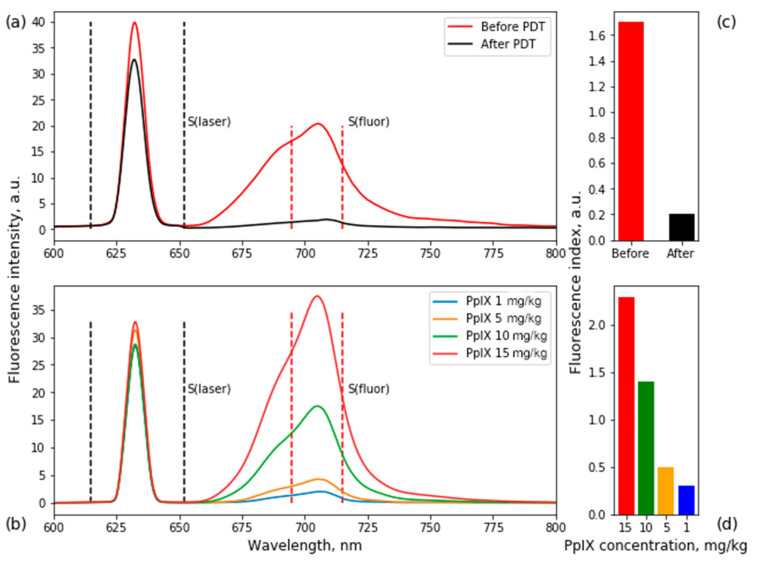
Fluorescence emission spectra of PpIX accumulated in tumor tissue before and after PDT (**a**), fluorescence spectra of optical phantoms with PpIX (**b**). Histograms of PpIX fluorescence index in tissues (**c**) and phantoms (**d**). Fluorescence index = S(fluor)/S(laser). Correlating fluorescence indices, PpIX concentrations before and after PDT were 12 and 0.7 mg/kg, respectively.

**Figure 4 bioengineering-09-00104-f004:**
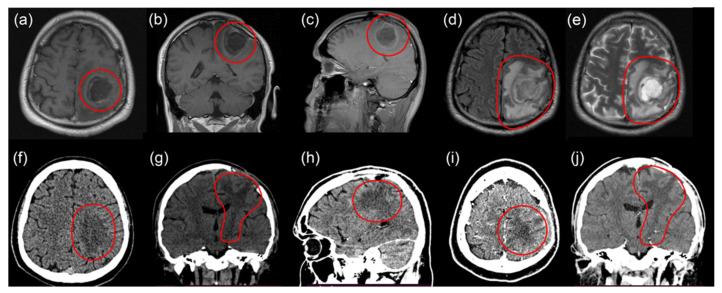
MRI images before third surgery: T1 axial (**a**), coronal (**b**), and sagittal (**c**) projections with contrast enhancement; T2 flair axial (**d**) and T2 axial (**e**) projections. CT images 2 months after third surgery: axial (**f**) and coronal (**g**) views. Ischemia of the left parietal lobe. CT images with contrast enhancement 2 months after third surgery: sagittal (**h**), axial (**i**), and coronal (**j**) views. No recurrent tumor and no contrast uptake in the tumor bed.

**Figure 5 bioengineering-09-00104-f005:**
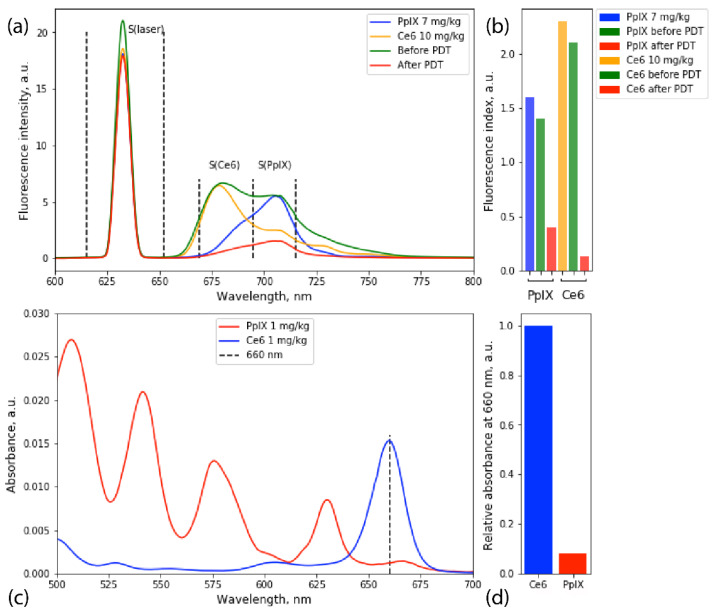
Fluorescence emission spectra of PpIX and Ce6 recorded before and after PDT and fluorescence spectra of prepared optical phantoms (**a**). Fluorescence indices histogram of PpIX and Ce6 in optical phantoms and accumulated in tumor tissue before and after PDT. Fluorescence index for PpIX and Ce6: S(Ce6 or PpIX)/S(laser). Correlating fluorescence indices, PpIX concentrations before and after PDT were 6 and 1.4 mg/kg, respectively; for Ce6, 9 and 0.6 mg/kg, respectively (**b**). Absorption spectra of Ce6 and PpIX at 1 mg/kg concentration (**c**). Relative absorbance of Ce6 and PpIX at 660 nm (**d**).

## Data Availability

The authors confirm that the data supporting the findings of this study are available within the article.

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
