# Peer review of "The Combined Use of 5-ALA and Chlorin e6 Photosensitizers for Fluorescence-Guided Resection and Photodynamic Therapy under Neurophysiological Control for Recurrent Glioblastoma in the Functional Motor Area after Ineffective Use of 5-ALA: Preliminary Results"

_bioengineering, 2022, doi:10.3390/bioengineering9030104_

Round 1
Reviewer 1 Report
This report describes the ability of a two-photosensitizer combination to promote PDT efficacy for the eradication of a glioma. It is proposed that this combination targets both tumor cells and tumor vasculature. A similar claim was made in 1996 for another two-photosensitizer combination: EtNBS and BPD (Visudyne). [Photochem Photobiol 63, 229-237, 1996]. It was later discovered that the efficacy of this combination resulted not from the targeting of tumor + vasculature but from the targeting of mitochondria and lysosomes. This was shown to result from the ability of lysosomal photodamage to promote the apoptotic response to mitochondrial photodamage [Photochem Photobiol. 93, 609-612].
Protoporphyrin, derived from ALA, is known to localize initially in mitochondria. Localization of chlorin e6 appears to involve both mitochondria and lysosomes [Oncology Letters 9, 551-556, 2015]. So it is possible that the efficacy of the e6 + ALA combination results not from targeting of tumor + vasculature but from targeting of mitochondria and lysosomes. There is a report indicating localization of chlorin e6 in mitochondria but also in lysosomes [Oncology Letters 9,551-556, 2015]. Data shown in Fig. 5c does suggest that anti-vascular effects can also be expected from Ce6-induced photodamage. The alternative hypothesis could be noted in the report. Perhaps a combination of both effects can account for the improved efficacy of the two-photosensitizer combination.
It would be useful to incorporate the wavelength of excitation into the legend to Figs. 3 and 5. The wavelength of excitation for the photosensitizer combination is specified to be 660 nm. I suggest showing the absorbance spectra of both protoporphyrin and chlorin e6 so as to indicate the relative absorbance of both agents at 660 nm. Protoporphyrin has a very low absorbance at this wavelength, while chlorin e6 has a very significant peak at this wavelength. This suggests that the major effect of the third intervention may have mainly involved chlorin e6-induced photodamage although the possibility of some contribution from protoporphyrin cannot be ruled out. Since there were no studies involving chlorin e6 alone, it is possible that the efficacy of the third intervention was related to chlorin e6 effects. It does appear that ALA alone was much less effective.
While this is clearly a preliminary report with a sample size of one, it is of considerable interest since treatment of glioma is often unsuccessful. Photodynamic therapy has not been especially effective although this could depend on inadequate light delivery. So while there are many issues not resolved, this does represent a rare case where photodynamic therapy had a successful outcome and does indicate a need for further exploration.
Reviewer 2 Report
The authors present a case study examining the use of two photosensitizers for fluorescence-guided resection and photodynamic therapy for the treatment of gliomas. Case studies such as the one reported in this manuscript are typically published in clinical journals. Due to the limited number of patients (one) and the very short follow-up (two months), it is not possible to make conclusions regarding the efficacy of the technique and thus, from a scientific point of view, the manuscript is unlikely to be of interest to readers of the Journal.
Line 47: the word “diapason” is not used in the correct context. Please re-write sentence for clarity.
Figure 4: I assume the T1 images in a, b and c were acquired with contrast? If so, please state in the figure caption.
Line 154: Please include the irradiation time.
Reviewer 3 Report
This case report shows the possibility of the combined use of 5-ALA and Chlorine photosensitizers for fluorescence-guided resection and photodynamic therapy of glioblastoma. In my opinion, this report can be so helpful for GBM therapy. I agree to publish this report in the journal. however, there are some typos that must be corrected. what is more, there are some complex and lengthy sentences that should be broken down to or even three short sentences which can be followed by readers.
Round 2
Reviewer 2 Report
The authors have not addressed my main concern. Due to the limited number of patients (one) and the very short follow-up (two months), it is not possible to make conclusions regarding the efficacy of the technique and thus, from a scientific point of view, the manuscript is unlikely to be of interest to readers of the Journal.
